# IMAGES AS WEIGHT MATRICES: SEQUENTIAL IMAGE GENERATION THROUGH SYNAPTIC LEARNING RULES

**Kazuki Irie**[1]   **Jürgen Schmidhuber**[1,2]
[1]The Swiss AI Lab, IDSIA, USI & SUPSI, Lugano, Switzerland
[2]AI Initiative, KAUST, Thuwal, Saudi Arabia
`{kazuki, juergen}@idsia.ch`

## ABSTRACT

Work on fast weight *programmers* has demonstrated the effectiveness of *key/value outer product-based learning rules* for sequentially generating a weight matrix (WM) of a neural net (NN) by another NN or itself. However, the weight generation steps are typically not visually interpretable by humans, because the contents stored in the WM of an NN are not. Here we apply the same principle to generate natural images. The resulting fast weight *painters* (FPAs) learn to execute sequences of delta learning rules to sequentially generate images as sums of outer products of self-invented keys and values, one rank at a time, as if each image was a WM of an NN. We train our FPAs in the generative adversarial networks framework, and evaluate on various image datasets. We show how these generic learning rules can generate images with respectable visual quality without any explicit inductive bias for images. While the performance largely lags behind the one of specialised state-of-the-art image generators, our approach allows for visualising how synaptic learning rules iteratively produce complex connection patterns, yielding human-interpretable meaningful images. Finally, we also show that an additional convolutional U-Net (now popular in diffusion models) at the output of an FPA can learn one-step "denoising" of FPA-generated images to enhance their quality. Our code is public.[1]

## 1 INTRODUCTION

A Fast Weight Programmer (Schmidhuber, 1991a; 1992) is a neural network (NN) that can learn to continually generate and rapidly modify the weight matrix (i.e., the program) of another NN in response to a stream of observations to solve the task at hand (reviewed in Sec. 2.1). At the heart of the weight generation process lies an expressive yet scalable parameterisation of update rules (or learning rules, or programming instructions) that iteratively modify the weight matrix to obtain any arbitrary weight patterns/programs suitable for solving the given task. Several recent works (Schlag et al., 2021a; Irie et al., 2021; 2022c;b) have demonstrated *outer products with the delta rule* (Widrow & Hoff, 1960; Schlag et al., 2021b) as an effective mechanism for weight generation. In particular, this has been shown to outperform the purely additive Hebbian update rule (Hebb, 1949) used in the Linear Transformers (Katharopoulos et al., 2020; Choromanski et al., 2021) in various settings including language modelling (Schlag et al., 2021a), time series prediction (Irie et al., 2022b), and reinforcement learning for playing video games (Irie et al., 2021). However, despite its intuitive equations—treating the fast weight matrix as a key/value associative memory—, the effective "actions" of these learning rules on the "contents" stored in the weight matrix still remain opaque, because in general the values stored in a weight matrix are not easily interpretable by humans.

Now what if we let a fast weight programmer generate a "weight matrix" that corresponds to some human-interpretable data? While outer product-based pattern generation may have a good inductive bias for generating a weight matrix of a linear layer[2], it can also be seen as a generic mechanism for iteratively generating any high dimensional data. So let us apply the same principle to generate and

---

[1]`https://github.com/IDSIA/fpainter`
[2]In a linear layer, the weight matrix is multiplied with an input vector to produce an output vector. Consequently, assuming that the output vector is used to compute some scalar loss function, the gradient of the loss w.r.t. weights is expressed as an outer product (between the input and the gradient of the loss w.r.t. the output).

incrementally refine natural images. We treat a colour image as three weight matrices representing synaptic connection weights of a fictive NN, and generate them iteratively through sequences of delta learning rules whose key/value patterns and learning rates are produced by an actual NN that we train. The resulting Fast Weight *Painters* (FPAs) learn to sequentially generate images, as sums of outer products, one rank at a time, through sequential applications of delta learning rules. Intuitively, the delta rule allows a painter to *look* into the currently generated image in a computationally efficient way, and to apply a *change* to the image at each painting step. We empirically observe that the delta rules largely improve the quality of the generated images compared to the purely additive outer product rules.

We train our FPAs in the framework of Generative Adversarial Networks (GAN; Goodfellow et al. (2014); Niemitalo (2010); Schmidhuber (1990); reviewed in Sec. 2.2). We evaluate our model on six standard image generation datasets (CelebA, LSUN-Church, Metfaces, AFHQ-Cat/Dog/Wild; all at the resolution of 64x64), and report both qualitative image quality as well as the commonly used Fréchet Inception Distance (FID) evaluation metric (Heusel et al., 2017). Performance is compared to the one of the state-of-the-art StyleGAN2 (Karras et al., 2020b;a) and the speed-optimised "light-weight" GAN (LightGAN; Liu et al. (2021)).

While the performance still largely lags behind the one of StyleGAN2, we show that our generic models can generate images of respectable visual quality without any explicit inductive bias for image processing (e.g., no convolution is used in the generator). This confirms and illustrates that generic learning rules can effectively produce complex weight patterns that, in our case, yield natural images in various domains. Importantly, we can visualise each step of such weight generation in the human-interpretable image domain. This is a unique feature of our work since learning rules are typically not visually meaningful to humans in the standard weight generation scenario—see the example shown in Figure 1 (weight generation for few-shot image classification).

Clearly, our goal is not to achieve the best possible image generator (for that, much better convolutional architectures exist). Instead, we use natural images to visually illustrate the behaviour of an NN that learns to execute sequences of learning rules. Nevertheless, it is also interesting to see how a convolutional NN can further improve the quality of FPA-generated images. For this purpose, we conduct an additional study where we add to the FPA's output, a now popular convolutional U-Net (Ronneberger et al., 2015; Salimans et al., 2017) used as the standard architecture (Ho et al., 2020; Song et al., 2021; Dhariwal & Nichol, 2021) for denoising diffusion models (Sohl-Dickstein et al., 2015). The image-to-image transforming U-Net learns (in this case) one-step "denoising" of FPA-generated images and effectively improves their quality.

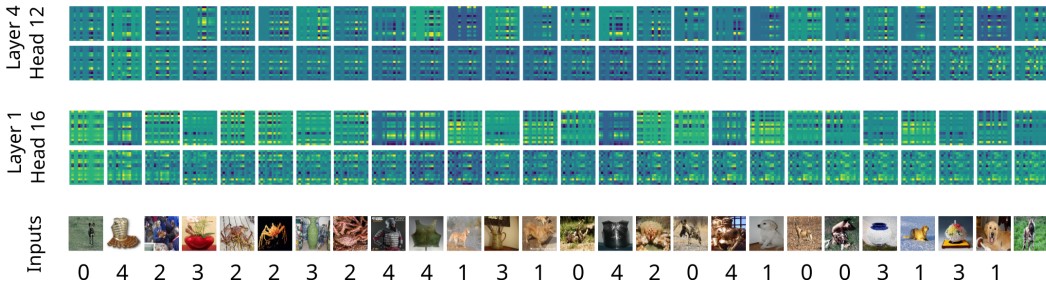

Figure 1: An illustration of the hardly human-interpretable standard weight generation process through sequences of delta rules in an FWP (DeltaNet) trained for 5-way 5-shot image classification on Mini-ImageNet (Vinyals et al., 2016; Ravi & Larochelle, 2017). The model is trained with the public code of Irie et al. (2022c) and achieves a test accuracy of 62.5%. The input to the model (shown at the bottom) is a sequence of images with their label (except for the last one to be predicted) processed from left to right, one image/label pair per step. The model has four layers with 16 heads each and a hidden layer size of 256. Each head generates a 16x16-dimensional fast weight matrix. Here we visualise weight generation of two heads: head '16' in layer 1 and head '12' in layer 4 as examples. In each case, the top row shows the rank-one update term (last term in Eq. 2), and the bottom row shows their cumulative sum, i.e., the fast weight matrix $W_t$ of Eq. 2, generated at the corresponding step.

## 2 BACKGROUND

Here we review the two background concepts that are essential to describe our approach in Sec. 3: Fast Weight Programmers (Sec. 2.1) and Generative Adversarial Networks (Sec. 2.2).

### 2.1 FAST WEIGHT PROGRAMMERS (FWPS)

A Fast Weight Programmer (FWP; Schmidhuber (1991a; 1992)) is a general-purpose auto-regressive sequence-processing NN originally proposed as an alternative to the standard recurrent NN (RNN). The system consists of two networks: the slow net, or the *programmer*[3] that learns (typically by gradient descent) to generate and rapidly modify a weight matrix (i.e., the program) of another net, the fast net, based on the input available at each time step. The context-sensitive fast weight matrix serves as short-term memory of this sequence processor. This concept has seen a recent revival due to its formal connection (Katharopoulos et al., 2020; Schlag et al., 2021a) to Transformers (Vaswani et al., 2017). In fact, Transformers with linearised attention (Katharopoulos et al., 2020) have a dual form (Aizerman et al., 1964; Irie et al., 2022a) that is an FWP with an outer product-based weight generation mechanism (Schmidhuber, 1991a; 1992). Let $t$, $d_{\text{key}}$, $d_{\text{in}}$, and $d_{\text{out}}$ denote positive integers. A notable example of such a model is the DeltaNet (Schlag et al., 2021a) that, at each time step $t$, transforms an input $\boldsymbol{x}_t \in \mathbb{R}^{d_{\text{in}}}$ into an output $\boldsymbol{y}_t \in \mathbb{R}^{d_{\text{out}}}$ while updating the fast weight matrix $\boldsymbol{W}_{t-1} \in \mathbb{R}^{d_{\text{out}} \times d_{\text{key}}}$ (starting from $\boldsymbol{W}_0 = 0$) as follows:

$$[\boldsymbol{q}_t, \boldsymbol{k}_t, \boldsymbol{v}_t, \beta_t] = \boldsymbol{W}_{\text{slow}} \boldsymbol{x}_t \tag{1}$$

$$\boldsymbol{W}_t = \boldsymbol{W}_{t-1} + \sigma(\beta_t)(\boldsymbol{v}_t - \boldsymbol{W}_{t-1}\phi(\boldsymbol{k}_t)) \otimes \phi(\boldsymbol{k}_t) \tag{2}$$

$$\boldsymbol{y}_t = \boldsymbol{W}_t \phi(\boldsymbol{q}_t) \tag{3}$$

where the slow net (Eq. 1; with a learnable weight matrix $\boldsymbol{W}_{\text{slow}} \in \mathbb{R}^{(2*d_{\text{key}}+d_{\text{out}}+1) \times d_{\text{in}}}$) generates query $\boldsymbol{q}_t \in \mathbb{R}^{d_{\text{key}}}$, key $\boldsymbol{k}_t \in \mathbb{R}^{d_{\text{key}}}$, value $\boldsymbol{v}_t \in \mathbb{R}^{d_{\text{out}}}$ vectors as well as a scalar $\beta_t \in \mathbb{R}$ (to which we apply the sigmoid function $\sigma$). $\phi$ denotes an element-wise activation function whose output elements are positive and sum up to one (we use softmax) that is crucial for stability (Schlag et al., 2021a). Eq. 2 corresponds to the rank-one update of the fast weight matrix, from $\boldsymbol{W}_{t-1}$ to $\boldsymbol{W}_t$, through the *delta learning rule* (Widrow & Hoff, 1960) where the slow net-generated patterns, $\boldsymbol{v}_t$, $\phi(\boldsymbol{k}_t)$, and $\sigma(\beta_t)$, play the role of *target*, *input*, and *learning rate* of the learning rule respectively. We note that if we replace this Eq. 2 with a purely additive Hebbian learning rule and set the learning rate to 1, i.e., $\boldsymbol{W}_t = \boldsymbol{W}_{t-1} + \boldsymbol{v}_t \otimes \phi(\boldsymbol{k}_t)$, we fall back to the standard Linear Transformer (Katharopoulos et al., 2020). In fact, some previous works consider other learning rules in the context of linear Transformers/FWP, e.g., a gated update rule (Peng et al., 2021) and the Oja learning rule (Oja, 1982; Irie et al., 2022b). Our main focus is on the delta rule above, but we'll also support this choice with an ablation study comparing it to the purely additive rule.

An example evolution of the fast weight matrix following delta rules of Eq. 2 is shown in Figure 1. As stated above, the visualisation itself does not provide useful information. The core idea of this work is to apply Eq. 2 to image generation, and conduct similar visualisation.

### 2.2 GENERATIVE ADVERSARIAL NETWORKS

The framework of Generative Adversarial Networks (GANs; Goodfellow et al. (2014); Niemitalo (2010)) trains two neural networks $G$ (as in generator) and $D$ (as in discriminator), given some dataset $\mathcal{R}$, by making them compete against each other. The GAN framework itself is a special instantiation (Schmidhuber, 2020) of the more general min-max game concept of Adversarial Artificial Curiosity (Schmidhuber, 1990; 1991b), and has applications across various modalities, e.g., to speech (Binkowski et al., 2020) and with limited success also to text (de Masson d'Autume et al., 2019), but here we focus on the standard image generation setting (Goodfellow et al., 2014). In what follows, let $c$, $h$, $w$, and $d$ denote positive integers. Given an input vector $\boldsymbol{z} \in \mathbb{R}^d$ whose elements are randomly sampled from the zero-mean unit-variance Gaussian distribution $\mathcal{N}(0,1)$, $G$ generates an image-like data $G(\boldsymbol{z}) \in \mathbb{R}^{c \times h \times w}$ with a height of $h$, a width of $w$, and $c$ channels (typically $c = 3$; a non-transparent colour image). $D$ takes an image-like input $\boldsymbol{X} \in \mathbb{R}^{c \times h \times w}$ and outputs a scalar

---

[3]Sometimes we refer to the slow net as the "fast weight programmer" by generally referring to it as an NN whose output is a weight matrix, e.g., see the first sentence of the introduction. However, without the forward computation of the fast net, the slow net alone is not a general purpose sequence processing system.

$D(\boldsymbol{X}) \in \mathbb{R}$ between 0 and 1. The input $\boldsymbol{X}$ is either a sample from the real image dataset $\mathcal{R}$ or it is a "fake" sample generated by $G$. The training objective of $D$ is to correctly classify these inputs as real (label '1') or fake (label '0'), while $G$ is trained to fool $D$, i.e., the error of $D$ is the gain of $G$.[4] The two networks are trained simultaneously with alternating parameter updates. The goal of this process is to obtain a $G$ capable of generating "fake" images indistinguishable from the real ones. In practice, there are several ways of specifying the exact form of the objective function. We refer to the corresponding description (Sec. 3.3) and the experimental section for further details.

Over the past decade, many papers have improved various aspects of GANs including the architecture (e.g., Karras et al. (2019; 2021)), loss function (e.g., Lim & Ye (2017)), data augmentation strategies (e.g., Zhao et al. (2020); Karras et al. (2020a), and even the evaluation metric (e.g., Heusel et al. (2017)) to achieve higher quality images at higher resolutions. In this work, our baselines are the state-of-the-art StyleGAN2 (Karras et al., 2020b;a) and a speed-optimised "light-weight" GAN (LightGAN; Liu et al. (2021)).

## 3 FAST WEIGHT "PAINTERS" (FPAS)

A Fast Weight Painter (FPA) is a generative model of images based on the weight generation process of outer product-based Fast Weight Programmers (FWP; Sec. 2.1; Eq. 2). Conceptually, such a model can be trained as a generator in the GAN framework (Sec. 2.2) or as a decoder in a Variational Auto-Encoder (VAE; Kingma & Welling (2014)). Here, we train it in the former setting that offers a rich set of accessible baselines. In the orginal FWPs, the slow net or the programmer's goal is to generate useful programs/weights for the fast net to solve a given problem. In the proposed FPAs under the GAN framework, the "slow net" or the *painter*'s objective is to generate images that maximise the "fast net"/discriminator/critic's prediction error.

Here we describe the main architecture of our FPA (Sec. 3.1), its extension through the U-Net (Sec. 3.2), and other GAN specifications (Sec. 3.3).

### 3.1 MAIN ARCHITECTURE

Like a typical generator in the GAN framework, an FPA is an NN that transforms a randomly-sampled latent noise vector into an image. Its general idea is to use a sequence processing NN to *decode* the input vector for a fixed number of steps, where in each step, we generate a key/value vector pair and a learning rate that are used in a synaptic learning rule to generate a rank-one update to the currently generated image (starting from 0). The final image thus corresponds to the sum of these update terms. The number of decoding (or *painting*) steps is a hyper-parameter of the model or the training setup.

In what follows, let $T$, $c$, $d_{\text{key}}$, $d_{\text{value}}$, $d_{\text{latent}}$, $d_{\text{in}}$, $d_{\text{hidden}}$, denote positive integers. Here we first provide an abstract overview of the building blocks of the FPA, followed by specific descriptions of each block. Given a random input vector $\boldsymbol{z} \in \mathbb{R}^{d_{\text{latent}}}$, and a number of painting steps $T$, an FPA generates an image $\boldsymbol{W} \in \mathbb{R}^{c \times d_{\text{value}} \times d_{\text{key}}}$ with $c$ channels, a height of $d_{\text{value}}$, and a width of $d_{\text{key}}$, through the following sequence of operations:

$$
\begin{align}
(\boldsymbol{x}_1, ..., \boldsymbol{x}_T) &= \text{InputGenerator}(\boldsymbol{z}) \tag{4} \\
(\boldsymbol{h}_1, ..., \boldsymbol{h}_T) &= \text{SequenceProcessor}(\boldsymbol{x}_1, ..., \boldsymbol{x}_T) \tag{5} \\
\boldsymbol{W}_t &= \text{UpdateNet}(\boldsymbol{W}_{t-1}, \boldsymbol{h}_t) \text{ for } t \in \{1, ..., T\} \tag{6} \\
\boldsymbol{W} &= \boldsymbol{W}_T \tag{7}
\end{align}
$$

where for $t \in \{1, ..., T\}$, $\boldsymbol{x}_t \in \mathbb{R}^{d_{\text{in}}}$, $\boldsymbol{h}_t \in \mathbb{R}^{d_{\text{hidden}}}$, and $\boldsymbol{W}_t \in \mathbb{R}^{c \times d_{\text{value}} \times d_{\text{key}}}$ with $\boldsymbol{W}_0 = 0$.

InputGenerator, SequenceProcessor, and UpdateNet denote the abstract blocks of operations described as follows:

**Input Generator.** As its name indicates, the role of InputGenerator is to transform the latent vector $\boldsymbol{z} \in \mathbb{R}^{d_{\text{latent}}}$ to a sequence of input vectors $(\boldsymbol{x}_1, ..., \boldsymbol{x}_T)$ with $\boldsymbol{x}_t \in \mathbb{R}^{d_{\text{in}}}$ for $t \in \{1, ..., T\}$, for the subsequent sequence processing NN, SequenceProcessor. In practice, we consider two variants.

---

[4]Despite the "adversarial" nature of this description, when $G$ is trained by gradient descent, it is nothing but the gradient feedback of $D$ through $D(G(z))$ that continually improves $G$. In this view, $D$ is "collaborative".

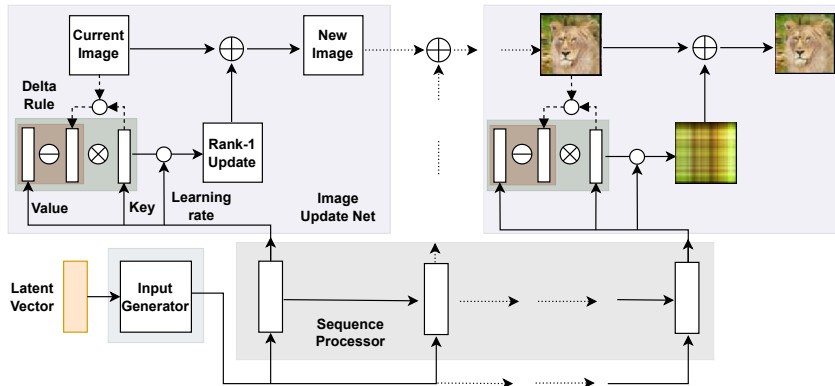

Figure 2: An illustration of the Fast Weight Painter architecture. The example is taken from the last painting step of our best FPA trained on the AFHQ-Wild dataset.

In the first variant (v1), the input generator does not do anything: it feeds the same latent vector to SequenceProcessor at every step, i.e., $\boldsymbol{x}_t = \boldsymbol{z}$ for all $t$ with $d_{\text{in}} = d_{\text{latent}}$. In the second variant (v2), InputGenerator is an NN that maps from an input vector of dimension $d_{\text{latent}}$ to an output vector of size $T * d_{\text{in}}$. The latter is split into $T$ vectors $\boldsymbol{x}_t$ ($t \in \{1, ..., T\}$) of size $d_{\text{in}}$ each. An obvious advantage of the first approach is that it scales independently of $T$, and it may also be rolled out for an arbitrary number of steps. In most datasets, however, we found the second variant to perform better, yielding more stable GAN training.

**Sequence Processor.** SequenceProcessor further processes the input vector sequence produced by InputGenerator. In our preliminary experiments, we found that auto-regressive processing and in particular RNNs are a good choice for this component (e.g., we did not manage to train any models successfully when the standard Transformer was used instead). We use a multi-layer long short-term memory (LSTM; Hochreiter & Schmidhuber (1997)) RNN in all of our models. We also allocate a separate NN that maps the latent vector $\boldsymbol{z}$ to the initial hidden states of each RNN layer (omitted for clarity in Figure 2).

**Image Update Network.** The actual painting finally emerges in the image update network, UpdateNet. The role of UpdateNet is to apply modifications to the current image $\boldsymbol{W}_{t-1}$ (starting from $\boldsymbol{W}_0 = 0$) and obtain a new image $\boldsymbol{W}_t$, through an application of the delta learning rule. At each time step $t$, the input $\boldsymbol{h}_t \in \mathbb{R}^{d_{\text{hidden}}}$ is projected to key $\boldsymbol{k}_t \in \mathbb{R}^{c*d_{\text{key}}}$, value $\boldsymbol{v}_t \in \mathbb{R}^{c*d_{\text{value}}}$, and learning rate $\beta_t \in \mathbb{R}^c$ vectors by using a learnable weight matrix $\boldsymbol{W}_{\text{slow}} \in \mathbb{R}^{c*(d_{\text{key}}+d_{\text{value}}+1) \times d_{\text{hidden}}}$

$$[\boldsymbol{k}_t, \boldsymbol{v}_t, \beta_t] = \boldsymbol{W}_{\text{slow}} \boldsymbol{h}_t \tag{8}$$

Similarly to multi-head attention in Transformers (Vaswani et al., 2017), the generated vectors are split into $c$ sub-vectors, one for each channel, i.e., for each painting step $t \in \{1, ..., T\}$, we have $[\boldsymbol{k}_t^1, ..., \boldsymbol{k}_t^c] = \boldsymbol{k}_t$ for keys, $[\boldsymbol{v}_t^1, ..., \boldsymbol{v}_t^c] = \boldsymbol{v}_t$ for values, and $[\beta_t^1, ..., \beta_t^c] = \beta_t$ for learning rates, where $\boldsymbol{k}_t^i \in \mathbb{R}^{d_{\text{key}}}$, $\boldsymbol{v}_t^i \in \mathbb{R}^{d_{\text{value}}}$, and $\beta_t^i \in \mathbb{R}$ for $i \in \{1, ..., c\}$ denoting the channel index.

Finally, for each step $t \in \{1, ..., T\}$ and for each channel index $i \in \{1, ..., c\}$, the corresponding image channel $\boldsymbol{W}_t^i \in \mathbb{R}^{d_{\text{value}} \times d_{\text{key}}}$ is updated through the delta update rule (same as in Eq. 2):

$$\boldsymbol{W}_t^i = \boldsymbol{W}_{t-1}^i + \sigma(\beta_t^i)(\boldsymbol{v}_t^i - \boldsymbol{W}_{t-1}^i \phi(\boldsymbol{k}_t^i)) \otimes \phi(\boldsymbol{k}_t^i) \tag{9}$$

Here each channel of the image is effectively treated as a "weight matrix," and "painting" is based on synaptic learning rules. The output image is finally obtained from the final step $t = T$ as $\boldsymbol{W} = [\boldsymbol{W}_T^1, ..., \boldsymbol{W}_T^c] \in \mathbb{R}^{c \times d_{\text{value}} \times d_{\text{key}}}$ (to which we apply $\tanh$). We'll visualise the iterative generation process of Eq. 9 in the experimental section.

An overview of this architecture is depicted in Figure 2.

### 3.2 OPTIONAL FINAL U-NET REFINEMENT STEP

Experimentally, we find the process described above alone can generate reasonably fine looking images *without* any explicit inductive bias for images. However, the resulting evaluation scores are worse than those of the baseline methods based on convolutional architectures. This motivates us to further investigate how the quality of images generated by an FPA can be improved by an additional convolutional component. For this purpose, we use the image-to-image transforming U-Net architecture (Ronneberger et al., 2015; Salimans et al., 2017), the core architecture (Ho et al., 2020; Song et al., 2021; Dhariwal & Nichol, 2021) of the now popular denoising diffusion models (Sohl-Dickstein et al., 2015). We apply the U-Net to the output of the FPA, i.e., after Eq. 7, it transforms the image $\boldsymbol{W} = \boldsymbol{W}_T \in \mathbb{R}^{c \times d_{\text{value}} \times d_{\text{key}}}$ into another image of the same size $\boldsymbol{W}' \in \mathbb{R}^{c \times d_{\text{value}} \times d_{\text{key}}}$:

$$\boldsymbol{W}' \quad = \quad \text{UNet}(\boldsymbol{W}_T) \tag{10}$$

From the U-Net's perspective, the output of the FPA is a "noisy" image. Its operation can be viewed as a one-step "denoising." This process is depicted in Figure 3. In practice, we append this U-Net to the output of a pre-trained FPA, and train the resulting model in the GAN framework, while only updating parameters of the U-Net. We also discuss end-to-end training in Appendix C.1. However, in the main setting, all our U-Net models are trained with a pre-trained frozen FPA. In the experimental section, we'll show that such a U-Net can effectively improve the quality of FPA-generated images.

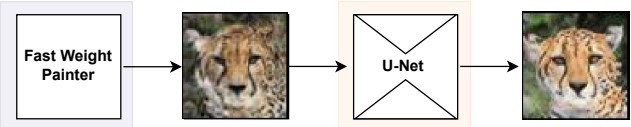

Figure 3: An illustration of the extra U-Net-based refinement. The example is generated by our best FPA/U-Net trained on the AFHQ-Wild dataset.

### 3.3 DISCRIMINATOR ARCHITECTURE AND OTHER SPECIFICATIONS

We adopt the training configurations and the discriminator architecture of the LightGAN (Liu et al., 2021), i.e., we replace its generator by our FPA. In our preliminary study, we also tried the StyleGAN2 setting, however, we did not observe any obvious benefit. In the LightGAN framework, in addition to the main objective function of the hinge loss (Lim & Ye, 2017; Tran et al., 2017), the discriminator is regularised with an auxiliary image reconstruction loss at two different resolutions (for that two "simple" image decoders are added to the discriminator). For details, we refer to Appendix B.3.

## 4 EXPERIMENTS

### 4.1 BENCHMARKING ON THE STANDARD DATASETS

We start with evaluating how well the proposed FPA performs as an image generator without explicit inductive bias for images. For this, we train our model on six datasets: CelebA (Liu et al., 2015), LSUN Church (Yu et al., 2015), Animal Faces HQ (AFHQ) Cat/Dog/Wild (Choi et al., 2020), and MetFaces (Karras et al., 2020a), all at the resolution of 64x64. No data augmentation is used as we want to keep the comparison as simple as possible. For details of the datasets, we refer to Appendix B.1. As a first set of experiments, we set the number of generation steps for FPAs to $T = 64$ (we'll vary this number in Sec. 4.3); such that the output images can be of full rank. We provide all hyper-parameters in Appendix B.2 and discuss training/generation speed in Appendix C.2. Following the standard practice, we compute the FID using $50\,\text{K}$ sampled images and all real images (for further discussion on the FID computation, we refer to Appendix B.4). Table 1 shows the FID scores. We first observe that the state-of-the-art StyleGAN2 outperforms our FPA by a large margin. The performance gap is the smallest in the case of the small dataset, MetFaces, while for larger datasets, CelebA and LSUN-Church, the gap is large. At the same time, the reasonable FID values show that the FPA is quite successful. Qualitatively, we observe that the FPA can produce images with a respectable quality, even though we also observe that the StyleGAN2 tends to generate fine-looking

Table 1: FID scores. The resolution is 64x64 for all datasets. No data augmentation is used. The StyleGAN2 models are trained using the official public implementation.

| Model | CelebA | MetFaces | AFHQ Cat | Dog | Wild | LSUN Church |
|---|---|---|---|---|---|---|
| StyleGAN2 | **1.7** | **17.2** | 7.5 | **11.7** | **5.2** | **2.8** |
| LightGAN | 3.4 | 26.4 | 7.9 | 16.8 | 10.2 | 5.1 |
| FPA | 18.3 | 36.3 | 17.1 | 45.3 | 20.2 | 42.8 |
| + U-Net | 3.7 | 24.5 | **6.8** | 19.9 | 9.4 | 5.2 |

images more consistently, and with a higher diversity. Figure 4 displays the curated output images generated by various models. By looking at these examples, it is hard to guess that they are generated as sums of outer products through sequences of synaptic learning rules. The visualisation confirms the FPA's respectable performance.

In Table 1, we also observe that the extra convolutional U-Net (Sec. 3.2) largely improves the quality of the images generated by the FPA. Its performance is comparable to that of the LightGAN baseline across all datasets. For further discussion of the UNet's effect, we refer to Appendix C.1.

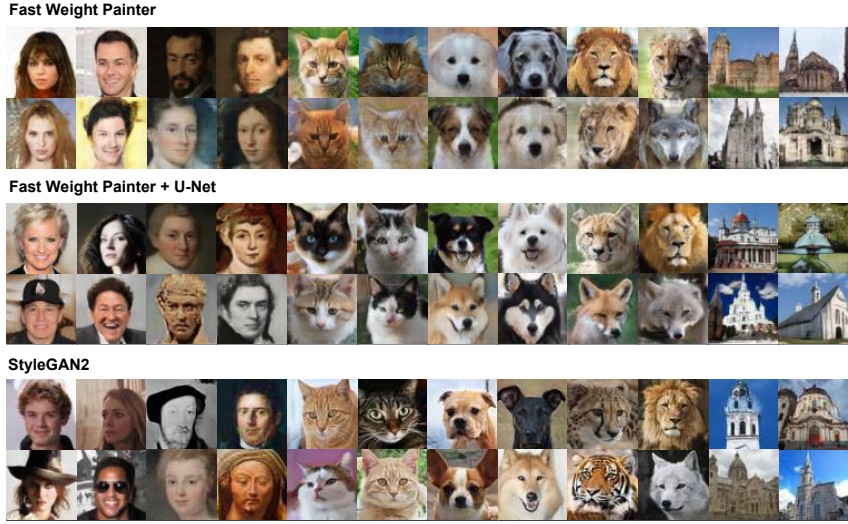

Figure 4: Curated image samples generated by different models specified above the images. Four images are shown for each dataset, from left to right: CelebA, MetFaces, AFHQ Cat, Dog, Wild, and LSUN Church. All at the resolution of 64x64.

## 4.2 VISUALISING THE ITERATIVE GENERATION PROCESS

How are images like those shown in Figure 4 generated iteratively as sums of outer products by the FPA? Here we visualise this iterative generation process. Generally speaking, we found almost all examples interesting to visualise. We show a first set of examples in Figure 5. As we noted above, all FPAs in Table 1/Figure 4 use $T = 64$ painting steps. The first thing we observe is that for many steps, the "key" is almost one-hot (which is encouraged by the softmax), i.e., the part of the image is generated almost column-wise. For other parts, such as generation/refinement of the background (e.g., steps 57-64 in AFHQ-Wild or steps 1-16 in MetFaces), rank-1 update covers a large region of the image. Generally we can recognise the "intended action" of each learning rule step on the image (e.g., adding the extra colour in the background in steps 59-64 in LSUN Church, or drawing a draft of the animal face in steps 7-34 in AFHQ-Wild). We discuss many more examples in Appendix A.

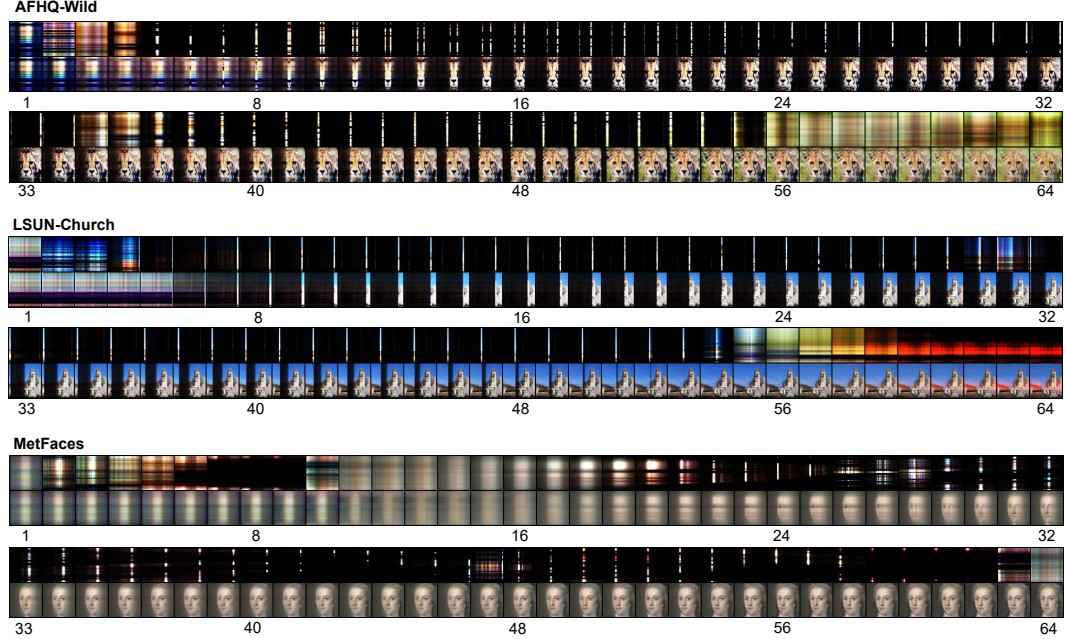

Figure 5: Example painting steps for the FPAs of Table 1. All images have the resolution of 64x64. In each example, the generation steps are shown from left to right. The numbers below images indicate the step. Each row has two mini-rows showing two sequences of images. The top mini-row shows the rank-1 update generated at the corresponding step, while the bottom mini-row shows the cumulative sum thereof, i.e., the currently generated image (Eq. 9). For visualisation, the image at each step is normalised by the norm of the final image, and we apply $\tanh$. We also note that the colour scaling is specific to each plot (in consequence, we observe some cases like the one we see in step 31 of LSUN-Church above where the effect of the rank-1 update is not visible once added to the image).

## 4.3 Ablation Studies

**Varying Number of Painting Steps.** In all examples above, we train FPAs with a number of painting steps $T = 64$ such that the 64x64 output can be of full rank. Here we study FPAs with reduced numbers of steps. The task should remain feasible, as natural images typically keep looking good (at least to human eyes, to some extent) under low-rank approximations (Andrews & Patterson, 1976). We select the model configuration that achieves the best FID with $T = 64$, and train the same model with fewer steps $\{8, 16, 32\}$. Table 2 shows the FIDs. Globally, we find that fewer steps

Table 2: FID scores of FPAs for various numbers of painting steps. The resolution is 64x64 for all datasets. The numbers for $T = 64$ are copied from Table 1.

| Steps $T$ | MetFaces | AFHQ | | |
| --- | --- | --- | --- | --- |
| | | Cat | Dog | Wild |
| 64 | 36.3 | 17.1 | 45.3 | 20.2 |
| 32 | 43.6 | 25.9 | 69.5 | 31.9 |
| 16 | 53.8 | 86.6 | 165.8 | 220.5 |
| 8 | 80.1 | 164.4 | 220.1 | 215.2 |

tend to greatly hurt performance. An exception is MetFaces (and AFHQ-Cat to some extent) where the degradation is much smaller. Figure 6 shows examples of 16-step generation of 64x64-resolution images for these two datasets, where we observe that FPAs find and exploit symmetries (for AFHQ-Cat) and other regularities (for MetFaces) as shortcuts for low rank/complexity image generation.

**Choice of Learning Rules.** As mentioned in Sec. 2.1, the delta rule is not the only way of parameterising the weight update (Eq. 9). However, we experimentally observe that both the purely additive rule and the Oja rule underperform the delta rule. This is in line with previous works on

FWPs (see introduction). The best FID we obtain on the CelebA using the purely additive rules is above 80, much worse than 18.3 obtained by the delta rule (Table 1). With the Oja rule, we did not manage to train any reasonable model on CelebA in our settings.

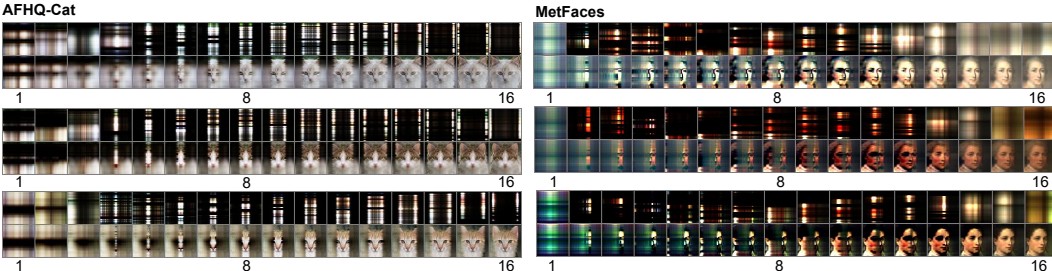

Figure 6: Examples of 16-step generation of 64x64 images by the FPA.

## 5    DISCUSSION

**Limitations.**    We have shown how the concept of FWPs can be applied to image generation, to visualise sequences of NN-controlled learning rules that produce natural images. While this contributes to the study of learning rules in general, we note that our visualisations are tailored to the image generation task: mapping a random vector to a sequence of images, where only the final image (their sum) is "evaluated." In many FWP use cases, however, the generated WMs are queried/used at every time step to solve some sequence processing task. What the learning rules actually do in such scenarios remains opaque. Also, from the perspective of image generation methods, there are many aspects that we do not investigate here. For example, we directly use the convolutional LightGAN discriminator. However, since the FPA is non-convolutional, there may be alternative architectures with better feedback/gradient properties for training FPAs. Also, our experiments are limited to images with a resolution of 64x64. Increasing the resolution is typically not straightforward (Karras et al., 2018) but outside the scope of this work.

**Diffusion models.**    We use the GAN framework (Sec. 2.2) to train our FPA, and mention the alternative of using VAEs. At first glance, it also seems attractive to use rank-1 noise as an efficient alternative to the expensive U-Net of diffusion models (Sohl-Dickstein et al., 2015). Unfortunately, unlike in the standard Gaussian case (Feller, 1949), if the forward process is based on rank-1 noises (e.g., obtained as outer products of two random Gaussian noise vectors), we have no guarantee that the reversal process can be parameterised in the same way using an outer product. Nevertheless, generally speaking, an exchange of ideas between the fields of image and NN weight generation/processing may stimulate both research domains. An example is the use of discrete cosine transform (DCT) to parameterise a WM of an NN (e.g., Koutník et al. (2010a;b); van Steenkiste et al. (2016); Irie & Schmidhuber (2021)).

**Other Perspectives.**    We saw that the same generic computational mechanism can be used for both fast weight programming and image generation (painting). From a cognitive science perspective, it may be interesting to compare painting and programming as sequential processes.

## 6    CONCLUSION

We apply the NN weight generation principles of Fast Weight Programmers (FWPs) to the task of image generation. The resulting Fast Weight Painters (FPAs) effectively learn to generate weight matrices looking like natural images, through the execution of sequences of NN-controlled learning rules applied to self-invented learning patterns. This allows us to visualise the iterative FWP process in six different image domains interpretable by humans. While this method is certainly not the best approach for image generation, our results clearly demonstrate/illustrate/visualise how an NN can learn to control sequences of synaptic weight learning rules in a goal-directed way to generate complex and meaningful weight patterns.

## ACKNOWLEDGEMENTS

We thank Róbert Csordás for helpful suggestions on Figure 2. This research was partially funded by ERC Advanced grant no: 742870, project AlgoRNN, and by Swiss National Science Foundation grant no: 200021_192356, project NEUSYM. We are thankful for hardware donations from NVIDIA and IBM. The resources used for this work were partially provided by Swiss National Supercomputing Centre (CSCS) project s1145 and s1154.

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

## A  MORE VISUALISATION EXAMPLES

As a continuation of Sec. 4.2, we provide further visualisations to illustrate the image generation process of FPAs.

Figure 7 shows some interesting effects we observe on CelebA. In Example 1 (top), from step 1 to 16, the image is column-wise generated from the left. Then from step 18, the generation of the right part of the face starts from the right. Around step 40 and 46, we see that the separately generated left and right parts of the face do not fit each other. But then the FPA fixes this in three steps from step 46 to 48 that "harmonise" the image. Example 2 (bottom) shows another similar example.

Figure 8 illustrates another typical behaviour we observe on both CelebA and AFHQ-Dog. In the CelebA example, we can see that the face is already generated around step 48. The rest of the steps are used to add extra hair. Similarly in the AFHQ-Dog example, step-48 image is already a fine-looking dog. An extra body is painted (rather unsuccessfully) from the right in the remaining steps.

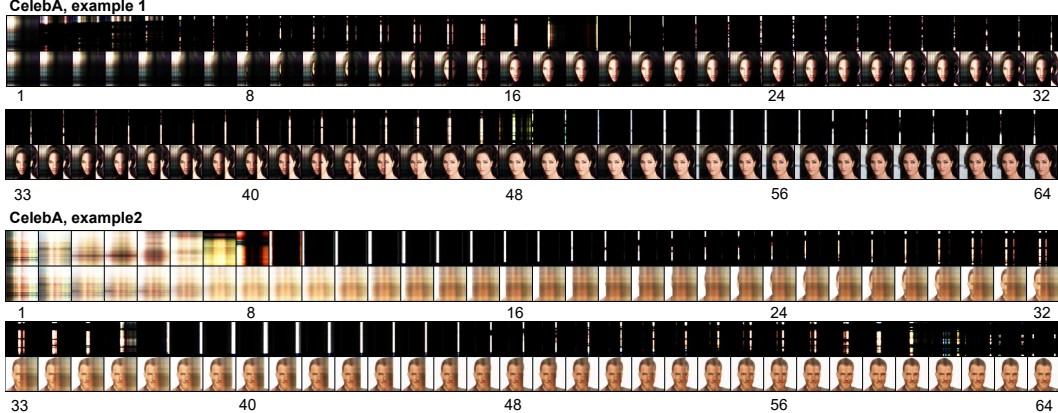

Figure 7: Example painting steps of the FPAs of Table 1 trained on CelebA at the resolution of 64x64. For details about the visualisation process, we refer to the caption of Figure 5.

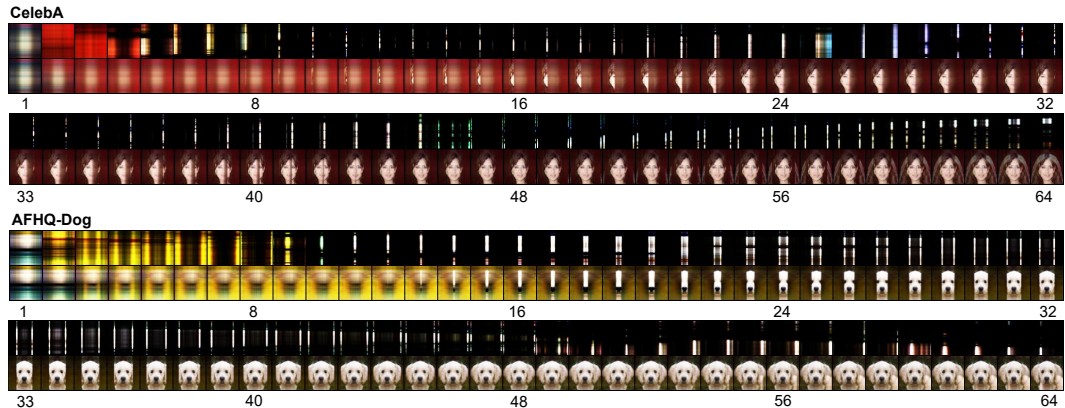

Figure 8: Example painting steps of the FPAs of Table 1 trained on CelebA and AFHQ-Dog at the resolution of 64x64. For details about the visualisation process, we refer to the caption of Figure 5 ( in particular, in step 25 of CelebA we observe an effect that is similar to step 31 of LSUN Church in Figure 5).

## B  Experimental Details

### B.1  Datasets

We use six standard benchmark datasets for image generation: CelebA (Liu et al., 2015), LSUN Church (Yu et al., 2015), Animal Faces HQ (AFHQ) Cat/Dog/Wild (Choi et al., 2020), and MetFaces (Karras et al., 2020a). The number of images are 203 K for CelebA, 126 K for LSUN Church, 5 K for each subsets of AFHQ, and 1336 for the smallest MetFaces. For further information about the datasets, we refer to the original papers.

### B.2  Hyper-Parameters

We conduct FPA tuning in terms of both hyper-parameters and model variations. Regarding the latter, we use three specifications: InputGenerator version v1 vs. v2 (see their descriptions in Sec. 3.1), input generator with or without $\tanh$, and with or without a linear layer that maps the latent vector to the RNN initial states (otherwise the initial states are zeros). We also train models with/without applying $\tanh$ to the generated image at the FPA output. Using the notations introduced in Sec. 3.1, the hyper-parameter search is conducted across the following ranges: $d_{\text{latent}} \in \{128, 256, 512\}$, $d_{\text{hidden}} \in \{1024, 2048, 4096\}$, and the number of RNN layers $L \in \{1, 2\}$. When InputGenerator is v2, we have $d_{\text{in}} \in \{8, 16, 32\}$ and furthermore, we add an extra linear layer between the input generator and the RNN with a dimension $d'_{\text{in}} \in \{128, 256, 512\}$. The batch size and learning rate are fixed to 20 and $2e^{-4}$ respectively. We compute the FID score every 5 K training steps to monitor the performance. We take the hyper-parameters achieving the best FID as the best configuration. Table 3 summarises the corresponding results. For MetFaces, a more parameter-efficient v2 variant (with 20 M parameters) achieves a FID of 40.5 that is close to 36.3 (Table 1) achieved by the best model. However, as the parameter count does not matter for our study, we strictly take the best model.

Table 3: Hyper-parameters of FPAs used for different datasets.

| Hyper-Parameter | CelebA | MetFaces | AFHQ | | | LSUN |
| | | | Cat | Dog | Wild | Church |
| --- | --- | --- | --- | --- | --- | --- |
| InputGenerator | v2 | v1 | v2 | v2 | v2 | v2 |
| Input Gen. $\tanh$ | No | - | No | No | No | Yes |
| Latent to RNN init | No | Yes | Yes | No | No | Yes |
| $d_{\text{latent}}$ | 128 | 512 | 256 | 256 | 512 | 512 |
| $d_{\text{in}}$ | 8 | - | 8 | 8 | 8 | 16 |
| $d'_{\text{in}}$ | 128 | - | 128 | 128 | 128 | 256 |
| $d_{\text{hidden}}$ | 1024 | 4096 | 1024 | 1024 | 1024 | 1024 |
| $L$ | 2 | 2 | 1 | 2 | 1 | 2 |
| Gen. Param. Count (M) | 14 | 220 | 6 | 14 | 6 | 17 |
| Total Param Count (M) | 26 | 232 | 18 | 26 | 19 | 29 |

### B.3  GAN Details & Implementation

As stated in Sec. 3.3, our GAN setting is based on that of the LightGAN by Liu et al. (2021). For architectural details of the discriminator, we refer to the original paper. To be more specific, our code is based on the unofficial public LightGAN implementation `https://github.com/lucidrains/lightweight-gan`. which is slightly different from the official implementation. In particular, the auxiliary reconstruction losses for the discriminator require 8x8 and 16x16 resolution images. In the original implementation, these are obtained directly from the intermediate layers of the generator, while in this unofficial implementation, they are obtained by scaling down the final output image of the generator by interpolation. The latter is the only compatible approach for FPAs since an FPA does not produce such intermediate images with small resolutions unlike the standard convolution based generators. For any further details, we refer to our public code. For the StyleGAN2

baseline, we use the official implementation of StyleGAN3 (Karras et al., 2021) that also supports StyleGAN2: `https://github.com/NVlabs/stylegan3`.

### B.4 FID COMPUTATION

While the Fréchet Inception Distance (FID; Heusel et al. (2017)) is a widely used metric for evaluating machine-generated images, it is sensitive to many details (Parmar et al., 2022). We also observe that it is crucial to use the same setting for all models consistently. We use the following procedure for all datasets and models (including the StyleGAN2 baseline). We store both the generated and resized real images in JPEG format, and use the `pytorch-fid` implementation of `https://github.com/mseitzer/pytorch-fid` to compute the FID. While the usage of PNG is generally recommended by Parmar et al. (2022), since consistency is all we need for the purpose of our study, and JPEG is more convenient for frequently monitoring FID scores during training (as JPEG images are faster to store, and take less disk space), we opt for it here.

Nevertheless, in Table 4, we also report FID scores computed using the generated and resized real images saved in PNG format. These are computed using the models from Table 1, i.e., the best model checkpoint found based on JPEG-based FID scores during training. All numbers grow beyond those in Table 1 including those of the baselines, but the general trend does not change. The FID scores reported in all other tables are computed using the JPEG format.

Table 4: FID scores using images in *PNG format*. The resolution is 64x64 for all datasets. No data augmentation is used.

| Model | CelebA | MetFaces | AFHQ | | | LSUN |
| | | | Cat | Dog | Wild | Church |
|---|---|---|---|---|---|---|
| StyleGAN2 | 2.7 | 24.8 | 11.7 | 17.4 | 7.0 | 9.8 |
| LightGAN | 8.8 | 47.2 | 13.0 | 27.9 | 14.0 | 9.6 |
| FPA | 33.6 | 73.7 | 23.4 | 71.6 | 28.6 | 64.7 |
| + U-Net | 6.9 | 43.1 | 9.8 | 30.5 | 12.6 | 8.5 |

## C EXTRA RESULTS & DISCUSSION

### C.1 DISCUSSION OF THE U-NET EXTENSION

**Visualising the U-Net's Effect.** In Sec. 4.1, we report the U-Net's quality improvements of FPA-generated images in terms of the FID metric. Here we further look into its effect on the actual images. Figure 9 displays some example images generated by FPA/U-Net models before and after the U-Net application. We observe that in general, the U-Net is successful at improving the quality of the FPA-generated images without completely ignoring the original images. However, we also observe examples where this is not the case, i.e., the U-Net generates almost completely new images as illustrated by the last row of MetFaces and AFHQ Dog examples.

**End-to-End Training.** As described in Sec. 3.2, we train the FPA/U-Net model using pre-trained FPA parameters. We also tried end-to-end training from scratch, but observed that the U-Net starts learning before the FPA can generate fine-looking images. As a result, the resulting U-Net ignores the FPA output, hence the FPA does not receive any useful gradients for learning to generate meaningful images, and the FPA remains a noise generator. We also tried to train it by providing both the FPA and U-Net outputs to the discriminator (and by stopping the gradient flow from the U-Net to the FPA). This alleviates the problem of the FPA remaining a noise generator, but is still not good enough to make the U-Net properly act as a denoising/refinement component of the FPA.

**Choice of the Pre-Trained FPA.** In general, we take the best performing standalone FPA as the pre-trained model to train a FPA/U-Net model (Sec. 3.2). In some cases, however, we find that the best standalone FPA model does not yield the best FPA/U-Net model. This is the case for AFHQ-Cat

and Dog as illustrated in Table 5. The FPA/U-Net model based on the best pre-trained FPA with an input generator of type v1 (Sec. 3.1) yields a slightly better FID than the one based on the best v2 variant, even though the trend is the opposite for the standalone FPA models. In Table 1 of the main text, we report the overall best FIDs.

Our U-Net implementation is based on the diffusion model implementation of `https://github.com/lucidrains/denoising-diffusion-pytorch`. We use three down/up-sampling ResNet blocks with a total of 8 M parameters.

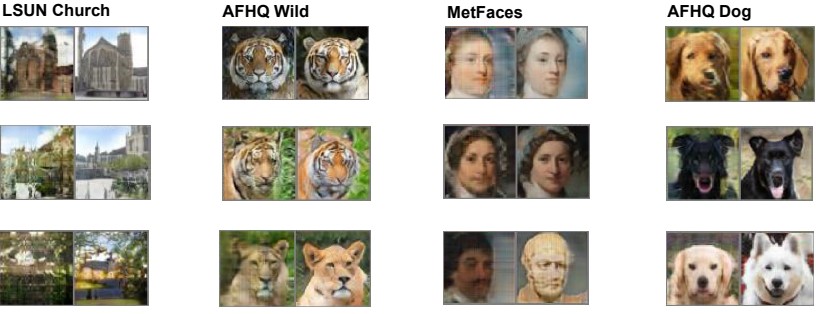

Figure 9: Example images generated by the FPA/U-Net before (left) and after (right) the U-Net.

Table 5: FID scores for the FPA/U-Net with various pre-trained FPA models. The resolution is 64x64 for all datasets. No data augmentation is used. v1/v2 indicates the type of the input generator described in Sec. 3.1.

| Model | AFHQ-Cat | | AFHQ-Dog | |
|---|---|---|---|---|
| | v1 | v2 | v1 | v2 |
| FPA | 30.8 | **17.1** | 49.3 | **45.3** |
| + U-Net | **6.8** | 7.8 | **19.9** | 22.9 |

## C.2 TRAINING AND GENERATION SPEED

While it is difficult to compare speed across different implementations, the generation speed of StyleGAN2 and the FPA are similar: about 55 images are generated every second for a batch size of one on a V100 GPU, while the LightGAN can generate 160 images per second in the same setting. The training/convergence speed of the FPA is similar to or better than that of the LightGAN on the relatively large datasets. For example, our FPA converges after about 150 K training steps on CelebA (vs. 285 K steps for the LightGAN), and 70 K steps on LSUN Church (similarly to the LightGAN) with a batch size of 20. The FPA/U-Net variant converges more slowly. For example on LSUN, it continues improving until 380 K steps in the same setting. On the small datasets, e.g., on AFHQ-Cat, the LightGAN converges faster (about 30 K steps) than the FPA (which continues improving until about 280 K steps) in the same setting. Any training run can be completed within one to three days on a single V100 GPU.

