# OpenReview forum: "Images as Weight Matrices: Sequential Image Generation Through Synaptic Learning Rules"
_ICLR.cc/2023/Conference — ICLR 2023 poster_

### Official Review · Reviewer_zop8 · 2022-10-24

**Confidence:** 4
**Correctness:** 4
**Technical Novelty And Significance:** 3
**Empirical Novelty And Significance:** 3
**Recommendation:** 6

**Clarity, Quality, Novelty And Reproducibility:**

While some figures can be further improved, this paper is presented in a very clear way supported by strong empirical results.
It is a novel idea on representing data in a domain-agnostic way.
I do not notice any reproducibility issues.

**Strength And Weaknesses:**

**Strength**

This paper demonstrates a very strong case for extending the fast weight idea in parameters space to other modalities like natural images.
While it is expected to see a lower performance compared to domain-specific methods, the authors show how to further close this gap by exploiting more inductive bias in the image domain.


**Weaknesses**

1. Figure 2 can be further improved by simply annotating everything in Eq 4-7 in the figure.
Now I'm very confused by the arrow pointing from 'Current image' to 'Value' and the one pointing from 'Key' to 'Current image'. What are they referring to?

2. While I understand this paper is just trying to illustrate a case of extending the idea of fast weight to more domains, more experiments certainly help to show how generic the proposed method is. Currently, there are only results with *image* + *GAN training*, while a quick experiment with other data domains like audio or training methods like VAE or diffusion can be a plus.



Minor points:

1.
In the Sequence Processor paragraph, *we did not manage to train any models successfully when the standard Transformer was used instead.* Does it mean transformers completely fail in this case? I don't see a clear reason why transformers cannot work at all here.

2. Based on the 2nd point in Weaknesses, I am very interested to see a synergy between the proposed method and diffusion, as both are iterative.



**Summary Of The Paper:**

This paper explores extending the idea of fast weight with outer-product based to human-interpretable domains beyond network weight space, and examines the idea with the example of natural image generation. The slow-weight network, now termed Fast Weight Painter now progressively refines the pixel values of a 3D image tensor with one rank-1 updating at a time step. Fair generation results can be observed without imposing any domain-specific inductive bias, and further performance improvements can be achieved by U-net-based one-step denoising.

**Summary Of The Review:**

This paper presents an interesting idea with strong but insufficient empirical results.
Please refer to Strength And Weaknesses for detailed comments.

---

> ### Author Response · Authors · 2022-11-08
> **Response to Reviewer zop8, part 1/2**
>
> We thank the reviewer for many positive comments.
>
> **Weaknesses**
>
> > *1. Figure 2 can be further improved by simply annotating everything in Eq 4-7 in the figure. Now I'm very confused by the arrow pointing from 'Current image' to 'Value' and the one pointing from 'Key' to 'Current image'. What are they referring to?*
>
> The explanation of the arrows is as follows.
> In the delta rule Eq. 2 (or 9), the key vector $k_t$ is first multiplied with the fast weight matrix $W_{t-1}$ (i.e, the current image) to retrieve the currently stored "value" vector $W_{t-1}*k_t$ (we omit the activation function).
> This is why there is an arrow going from "Key" to "Current image" (as they are multiplied).
> Then, the effective value vector used for the image update is obtained by subtracting the currently stored value vector $W_{t-1}*k_t$ (retrieved from the current image) from the newly generated value vector $v_t$ (i.e., $v_t - W_{t-1}*k_t$).
> This dependency is illustrated by the arrow going from "Current image" to "Value."
> But we fully agree with the reviewer that these arrows may be too abstract and confusing.
> We will improve Figure 2 by adding nodes representing multiplication and subtraction operations that directly reflect the corresponding equations as the reviewer suggests.
> We thank the reviewer for pointing this out.
>
> > *2. ... more experiments certainly help to show how generic the proposed method is. Currently, there are only results with image + GAN training, while a quick experiment with other data domains like audio or training methods like VAE or diffusion can be a plus.*
>
> We fully agree with the reviewer that experiments covering multiple image generation paradigms would strengthen our paper.
> In fact, at an early stage of this project, we did start our preliminary experiments with VAEs as a proof of concept that turned out to be successful.
> However, for proper evaluation, we need solid baselines (like StyleGAN2 and LightGAN used in this paper) that are fairly easily reproducible without requiring excessive computational resources.
> Not only typical VAEs lag behind GAN-trained generators in terms of image generation quality, we could not find well performing VAE baselines that can be trained fast enough with our academic computational resources.
> For this reason, we opted to focus on GANs for this first paper on FPAs.
> This is the full story behind our comment in the first paragraph of Sec. 3 *"Here, we train it in the former setting that offers a rich set of accessible baselines"* (where *"former"* refers to GANs).
> Conducting proper evaluation for VAEs (or on other modalities) would require efforts for almost another full paper.
> Thus, we rather leave it for future work.
> In fact, to the best of our knowledge, a conference paper covering both GAN and VAE settings with convincing experimental settings for both cases is rather uncommon.
>
> Regarding the diffusion models, we comment on it below.
>
> **Minor points**
>
> > *1. In the Sequence Processor paragraph, we did not manage to train any models successfully when the standard Transformer was used instead. Does it mean transformers completely fail in this case? I don't see a clear reason why transformers cannot work at all here.*
>
> Yes, the standard Transformers completely failed. Since we expected this comment, we included code for Transformers in our supplemental material (to support our claim that we did try them).
> Please note that the task here is not a standard sequence processing task.
> The inputs are random, and the sequence model only gets feedback on the final image generated at the last step.
> This means that the sequence processor needs to learn to decompose the task of image generation into multiple steps, without any explicit intermediate supervision.
> Recurrence (that encourages proper step-wise computation) seems to help in this scenario.
> We also recall that there are sequence processing tasks where standard Transformers fail, as recurrence is crucial.
> A good recent example is the code execution task of [zop8, 1].
> [zop8, 2] show that adding recurrence to the (linear) Transformer effectively makes them successful at this task.
>
> [zop8, 1] Fan et al. "Addressing some limitations of Transformers with feedback memory" (see Table 1, row "Algorithmic")
>
> [zop8, 2] Irie et al. "Going Beyond Linear Transformers with Recurrent Fast Weight Programmers" NeurIPS 2021. (See Table 2)

---

> > ### Author Response · Authors · 2022-11-08
> > **Response to Reviewer zop8, part 2/2**
> >
> > > *2. Based on the 2nd point in Weaknesses, I am very interested to see a synergy between the proposed method and diffusion, as both are iterative.*
> >
> > We actually do discuss this in the paper. Please see the second paragraph *"Diffusion models"* of Sec. 5 "Discussion."
> > The use of rank-one noises/updates seems very attractive as an alternative to the expensive U-Net based updates. However, rank-one noises obtained as outer products between Gaussian noise vectors are not Gaussian. Therefore, we have no theoretical support that the reversal can also be parameterised as rank-one updates.
> > Actually there are also some other engineering issues with the rank-one variant (e.g., it is not straightforward to efficiently sample noises of various levels, again due to the non-Gaussian property).
> > Therefore, we rather leave this for separate future work.
> >
> > **Summary**
> >
> > > *... interesting idea with strong but insufficient empirical results*
> >
> > Work presenting unconventional methods (like ours) tends to have empirical results limited to a few toy datasets such as MNIST or similar.
> > Here we did run experiments on six standard datasets.
> >
> > Overall, we believe that our response above should resolve all the main concerns raised by the reviewer. If you think that the paper should be accepted, please consider increasing the score. Thank you.

---

> > > ### Author Response · Authors · 2022-11-17
> > > **Friendly reminder**
> > >
> > > This is a friendly reminder that the discussion period ends tomorrow.
> > > If you find our response convincing, please consider increasing the score.
> > > Thank you!

---

### Official Review · Reviewer_81gc · 2022-10-26

**Confidence:** 3
**Clarity, Quality, Novelty And Reproducibility:** 1. The author has provided source cod…
**Correctness:** 3
**Technical Novelty And Significance:** 3
**Empirical Novelty And Significance:** 3
**Recommendation:** 5

**Strength And Weaknesses:**

Pros:
1. The paper is well-written.
2. The proposed method is technically sound.
3. It is novel to introduce fast weight programmer method into image generation task.

Cons:
1. The generation results is not comparable to SOTA image generation method, although I acknowledge this work is an attempt to introduce fast weight programmer into image generation task.

2. It is good to see the decomposition of image generation process step-by-step, however, the author didn't provide enough illustration or explanation for the visualization results.

3. Lack of high-resolution image generation results.

**Summary Of The Paper:**

This paper presents an image generation model called fast weight painters (FPA), applying the technique of fast weight programmer. The proposed method treats the generated image as the generated weight matrix.

1. This work seems to be one of the first attempts to apply fast weight programmer method to image generation task, which is novel.
2. The iterative optimizations of image generation also provide a good interpretability.

**Summary Of The Review:**

Overall, the proposed method looks novel to me, although not flawless as mention above. I would recommend to accept this paper.

UPDATE: After discussing with other reviewers and AC, I decide to lower the score. We agree the proposed method is novel and interesting. But as I mentioned the the review, the paper needs more explanation for the visualization results. And the author should also demonstrate why it is important to the community. We believe the paper could be more solid if the author could address these issues.

---

> ### Author Response · Authors · 2022-11-08
> **Response to Reviewer 81gc**
>
> We thank the reviewer for many positive comments.
>
> We skip the first point (on the performance lagging behind the SOTA) as the reviewer seems to accept it by acknowledging the purpose of our work.
>
> > *2. It is good to see the decomposition of image generation process step-by-step, however, the author didn't provide enough illustration or explanation for the visualization results.*
>
> The corresponding explanations are provided in Sec. 4.2.
> Generally speaking, we tried to keep the descriptions as succinct as possible by grouping statements that are valid across multiple examples or generation steps, instead of commenting every single step/example.
> We actually also provide more illustrations and explanations in Appendix A.
> Nevertheless, if the reviewer has any more specific requests, we'll be happy to address them.
>
> > *3. Lack of high-resolution image generation results.*
>
> We do discuss this in the second paragraph under "Limitations" in Sec. 5 (*"Also, the experiments conducted here are limited to..."*).
> Actually, we did run some experiments with higher resolutions, but we did not manage to obtain any satisfactory results using the same hyper-parameters used for the 64x64 case.
> Please note that the difficulty of scaling image generation methods to higher resolutions is not specific to our method.
> Even for the conventional convolution based GAN models, this has been a research question in multiple works (see e.g., [81gc, 1] and [81gc, 2]).
> Similarly, we feel that this is not something that we can easily fix in this first paper on FPAs.
> If the reviewer wishes, we'll be happy add more comments on this in the revision, but please note that the current text already does mention this limitation explicitly by citing [81gc, 2].
>
> [81gc, 1] Odena et al. Conditional Image Synthesis with Auxiliary Classifier GANs. ICML 2017
>
> [81gc, 2] Karras et al. Progressive Growing of GANs for Improved Quality, Stability, and Variation. ICLR 2018
>
> We hope that our response above convincingly addresses the reviewer's main concerns. If you think that the paper should be accepted, please consider increasing the score. Thank you.

---

> > ### Author Response · Authors · 2022-11-17
> > **Friendly reminder**
> >
> > This is a friendly reminder that the discussion period ends tomorrow.
> > If you find our response convincing, please consider increasing the score.
> > Thank you!

---

### Official Review · Reviewer_oERk · 2022-10-27

**Confidence:** 3
**Correctness:** 3
**Technical Novelty And Significance:** 2
**Empirical Novelty And Significance:** Not applicable
**Recommendation:** 6

**Clarity, Quality, Novelty And Reproducibility:**

The writing is clear,  and overall reproducibility and novelty should be fine.
More training details and network architecture would be appreciated.

**Strength And Weaknesses:**

**Strength**:
1. I like the core idea of this paper and feel it is quite interesting. Images are generated as the summation of a sequence of rank-one matrices following the delta learning rules, whose matrices are outer products of key/value invented from the neural network.
The self-invented value/key has subtle connections to transformers, and the iterative updating mechanism is intuitively related to diffusion models,  where adding new ranks is a kind of "denoising." I feel this paper may have the potential to inspire new image-generation methods.
2. The writing is good, and the paper is easy to follow.

**Weakness**
Although the core idea is cool, my concern is: the current paper doesn't provide enough intriguing results/contributions, which are listed below.
1. **Using image generation task to understand the weight generation process.** One motivation for this paper is to "visually illustrate the behavior of an NN that learns to execute sequences of learning rules." Visualizing weight generation sequence is hard to understand, so authors visualize image generation.
I am not convinced by this statement since image generation and weight generation are quite different, and it is unclear to me how much visualization/observation obtained from visualizing image generation could be transferred/generalized to weight generation.
Moreover, observations from visualization(Sec 4.2, Appendix A) are quite specific for each dataset, which doesn't reveal any genetic observations/ideas.
Some experiments, like summarising the genetic properties of the generation process and validating it on different domains, would support this claim. Otherwise, I personally think it is specific to image generation tasks.
2. I understand "Clearly, our goal is not to achieve the best possible image generator (for that, much better convolutional
architectures exist). Instead, we use natural images to visually illustrate the behavior of
an NN ..." But I feel more solid experiments/design exploration might improve this paper.
E.g., the current image generation results have a big gap to stylegan 2 (which is fine for me), and are limited to 64x64 resolution. I wonder, does it still work in 128x128 or even higher resolution?
What if inventing several key/value pairs at each step and calculating multi-rank matrices?
What if updating is conducted on each image patch instead of the whole image?
Why UNet's induct bias is so helpful?
It is not necessarily a weakness but adding more design exploration/analysis would be helpful to illustrate the power of the proposed method as an image generation method.


In summary, as a paper visualizing the generation process, weakness 1 would be my main concern. From the image generation perspective, I am worried about the current performance and experiments.

**Summary Of The Paper:**

Fast weight programming sequentially generates a weight matrix of a neural network from another neural network, which matrix is generated as the summation of outer products of self-invented keys and values.
This paper employs the idea of fast weight programming in the image generation task,  which results in fast weight painters (FPAs).
FPAs learn to generate images following a sequence of delta learning rules and adding rank-one components to existing images at each step.
The authors train FPAs in the GAN framework in 64x64 resolution and evaluate it on various image datasets.
While the performance largely lags behind the StyleGAN2,  authors claim this method allows for visualizing the iterative procedure of complex connections patterns in synaptic learning rules.

**Summary Of The Review:**

This paper is interesting, at least for me.
My major concern is: the observation from the image generation is dataset specific and not helpful in understanding the generic process.
As an image generation method, it is not powerful and well-designed enough.

---

> ### Author Response · Authors · 2022-11-08
> **Response to Reviewer oERk**
>
> We thank the reviewer for many positive comments on the originality and clarity of our work.
>
> **Weaknesses**
>
> > *1. Using image generation task to understand the weight generation process ... I am not convinced by this statement since image generation and weight generation are quite different, and it is unclear to me how much visualization/observation obtained from visualizing image generation could be transferred/generalized to weight generation.*
>
> Let us resolve an important misunderstanding.
> The reviewer wrote *"Using image generation task to understand the weight generation process."* This is not our goal, and we never claim that what we observe for image generation extrapolates to the weight generation case.
> On the contrary, we completely agree with the reviewer that this is certainly not the case,
> and we explicitly comment on this in our discussion Sec. 5 (under "Limitations"): *"While these
> illustrations contribute to the study of learning rules in general, we note that these visualisations are specific to the image generation task."*
>
> As the reviewer quotes, our goal is to *"visually illustrate the behavior of an NN that learns to execute sequences of learning rules."*
> We apply FWPs to image generation to demonstrate that the generic NN weight learning rules can effectively produce arbitrary complex patterns (we illustrate this through natural images),
> and to visualise these sequences of learning rules that specifically produce these images.
> While we believe we clearly state this in the introduction (fourth paragraph) and the conclusion, please let us know if the reviewer still finds this confusing.
>
> We believe this to be an original contribution to the study of NN learning rules, as no previous work has
> had good visualisation/illustration of sequences of learning rules for ANY tasks.
>
> Since the reviewer refers to this point as her/his major concern (*"weakness 1 would be my main concern"*), we hope that
> resolving this misunderstanding will significantly improve the reviewer's rating of this work.
>
> Now the second point under weaknesses is rather a collection of questions/suggestions (*"It is not necessarily a weakness"* in the reviewer's own words):
>
> > *I wonder, does it still work in 128x128 or even higher resolution?*
>
> Reviewer 81GC asked a similar question.
> Actually, we do discuss this in the second paragraph under "Limitations" in Sec. 5 (*"Also, the experiments conducted here are limited to..."*).
> In fact, we did run some experiments with higher resolutions, but we did not manage to obtain any satisfactory results using the same hyper-parameters used for the 64x64 case.
> Please note that the difficulty of scaling image generation methods to higher resolutions is not specific to our method.
> Even for the conventional convolution-based GAN models, this has been a research question in multiple works (see e.g., [oERk, 1] and [oERk, 2]).
> Similarly, we feel that this is not something that we can easily fix in this first paper on FPAs.
> If the reviewer wishes, we'll be happy add more comments on this in the revision, but please note that the current text already does mention this limitation explicitly by citing [oERk, 2].
>
> [oERk, 1] Odena et al. Conditional Image Synthesis with Auxiliary Classifier GANs. ICML 2017
>
> [oERk, 2] Karras et al. Progressive Growing of GANs for Improved Quality, Stability, and Variation. ICLR 2018
>
> > *What if inventing several key/value pairs at each step and calculating multi-rank matrices? What if updating is conducted on each image patch instead of the whole image?*
>
> We thank the reviewer for all these interesting ideas to extend this work. Indeed, we had thought about patch-wise generation using multiple heads at an early stage of this project.
> However, since these ideas are rather orthogonal to the main purpose of this work, we leave them to future work.
>
> > *Why UNet's induct bias is so helpful?*
>
> U-Net is a convolutional NN (and thus has inductive bias for images). Since it is the main architecture of the now popular diffusion models, it is not surprising that it works well as a generic image to image transformation block.
>
> We hope that our response resolves the reviewer's main concerns (in particular the misunderstanding regarding weakness 1).
> If that is the case, please consider increasing the score.
> Otherwise, please specifically explain the reasons for rating this work as 5 (and thus voting to reject this paper).
> Thank you.

---

> > ### Comment · Reviewer_oERk · 2022-11-16
> > **Response**
> >
> > Thanks for the feedback from the authors.
> >
> > I am still not fully convinced by this response. Let me re-state my concern here.
> >
> > My major concern is the contribution of this paper. It could be either "1. illustrate the behavior of an NN that learns to execute sequences of learning rules." or "2. a new image generation pipeline with the potential to contribute to the community". Either 1 or 2 is acceptable to me.
> >
> > I suggested several further modifications or "what if" things since I felt that it was hard to convince people this design could be a powerful generation technique from the current version and more experiments would reveal its potential advantages. Since the authors agree that the main target of this paper is 1 instead of image generation, then the title is confusing to me since it only talks about image generation.
> >
> >  For target 1, my major concern is: is this image generation example really helpful for people to understand the generic behavior of NN?
> > It makes sense to use image generation as a starting point, but this paper doesn't draw generic conclusions beyond image generation~(Please highlight them if I miss them).
> > For instance,
> > authors argued that **We apply FWPs to image generation to demonstrate that the generic NN weight learning rules can effectively produce arbitrary complex patterns (we illustrate this through natural images)**.
> > First, it is not immediately clear to me why natural images have arbitrary complex patterns since natural images are much smoother than neural network weights.
> > Second, it is hard to tell whether this observation is helpful for future studies.
> > Another example is the visualization of the generation procedure in Figure 5. *the “key” is almost one-hot (which is encouraged by the softmax).* At least the authors should show whether this observation holds in neural network weight generation to demonstrate the generic behaviors (I personally think it still holds since you are using softmax).
> >
> > All in all, I am worried that the current paper does not help understand generic behaviors beyond these image generations.
> > Further discussions are welcomed.

---

> > > ### Author Response · Authors · 2022-11-17
> > > **Re: Response, Reviewer oERk**
> > >
> > > We thank the reviewer for the response.
> > > > *the title is confusing to me since it only talks about image generation.*
> > >
> > > We do not fully agree with the reviewer’s claim that the current title of our paper *“only talks about image generation.”* The title contains other keywords such as “weight matrices” and “synaptic learning rules.” In fact, the originality of our work lies in the combination of these concepts with the image generation task, and we believe that this is rather well reflected in the title. We expect this title to attract researchers interested in NN learning rules, not *“only those interested in image generation.”*
> > >
> > > > *For target 1, my major concern is: is this image generation example really helpful for people to understand the generic behavior of NN?*
> > >
> > > No, we never claim that. This is the misunderstanding we tried to resolve in the first paragraph of our response: *“Let us resolve an important misunderstanding. … we never claim that what we observe for image generation extrapolates to the weight generation case.”* We do **not** intend to draw any **generic conclusion** (we explicitly state this in our limitation section 5, as we already pointed out in our response). We focus on the very specific case of natural images generation. Nevertheless, this allows us:
> > >
> > > (1) to show that learning rules can be used to generate anything other than NN weight matrices. This is what we mean by *“arbitrary complex patterns.”* But we would be happy to remove “complex” and write “arbitrary patterns” instead, since the reviewer is right to point out that images may not be “arbitrary complex.” We claim that this is a contribution to research on NN learning rules, since unlike any prior work, we do demonstrate that NN learning rules can be used for anything other than NN weight generation and thus, they are more **generic mechanisms** for data generation.
> > >
> > > (2) to illustrate (not *“understand”*) sequential processes of NN learning rules for this very specific case where the final matrix is a human interpretable image. We can not *“tell whether this observation is helpful for future studies,”* but we do see how these sequences of learning rules generate images. We do not think such processes/steps are obvious even for experts in NN learning rules, and therefore, we argue that such illustrations themselves have values.
> > >
> > > > *At least the authors should show whether this observation holds in neural network weight generation to demonstrate the generic behaviors (I personally think it still holds since you are using softmax).*
> > >
> > > In light of the clarification above, this question is (hopefully) not relevant anymore, but we can still answer for the sake of completeness. Please note that we can not even claim anything general among NN weight generation processes, since they might depend on the task. What we see in Figure 1 (NN weight generation for few shot image classification) can be very different from weight generation in FWPs trained for language modelling (this is also true for activations in other generic NNs like LSTM gates). Please note, however, that the crucial difference between Figure 1 and other Figures showing image generation is that, in Figure 1 we can not even tell if the final weight matrices are meaningful or not, and in this sense, Figure 1 is not even a meaningful illustration.
> > >
> > > > *Further discussions are welcomed.*
> > >
> > > We appreciate it, thank you!
> > >
> > > We hope that our response finally resolves the reviewer's main concern.

---

> > > > ### Comment · Reviewer_oERk · 2022-11-18
> > > > **Re: Response**
> > > >
> > > > Thanks for the response of the authors and I think it resolves all misunderstandings.
> > > >
> > > > The core contribution of this paper is to show that the sequential learning rule could be utilized to generate images.
> > > > I personally feel this contribution may not be that strong, but I am ok with accepting this paper.

---

> > > > > ### Author Response · Authors · 2022-11-18
> > > > > **Thank you**
> > > > >
> > > > > Thank you very much for your response and the updated score.

---

### Official Review · Reviewer_wMfX · 2022-10-31

**Confidence:** 4
**Correctness:** 3
**Technical Novelty And Significance:** 2
**Empirical Novelty And Significance:** 2
**Recommendation:** 8

**Clarity, Quality, Novelty And Reproducibility:**

*Clarity*

The paper is mostly clear to read. But would appreciate a more vivid view of the FPA generator architecture and the role of a third NN to map z to initial hidden states of RNN layers of the slow NN.


*Reproducibility*

The work builds upon the existing architecture and has given links to the versions of the implementations used in the work. The work should be fairly reproducible given the code for FPA generator is released

*Novelty and Originality* (repeated from above)

The work is interesting in it being one of the few works that draw similarities between images and weight matrices that are generated by another neural net. The recent of this type being the class of Implicit Neural Representations (INR) [1 ]. The work also is a a slightly novel extension to the existing literature on FWP to image generation.




**Strength And Weaknesses:**

*Novelty and Originality*

The work is interesting in it being one of the few works that draw similarities between images and weight matrices that are generated by another neural net. The recent of this type being the class of Implicit Neural Representations (INR) [1 ]. The work also is a a slightly novel extension to the existing literature on FWP to image generation.


*Methodology*

The authors claims that image generation is not the best of the strengths of FPA, leaving the visualization of weight updates as the central purpose of the work. Given the poor generation quality of FPA, my question is if the  performance vs interpretability/explainability tradeoff presented in the work is worth a shot. Can the method be adapted to exercise control over generations by manipulating the low-rank updates, ref [4] or restrict the low-rank updates to find discriminating components, ref [5], instead of a rank based updation approach.

Why shouldn't  one compare FPA with recurrent painting techniques like Draw [2].

The authors posit that FPA does not include any explicit inductive bias as noted in page 2, "*we show that our
generic models can generate images of respectable visual quality without any explicit inductive
bias for image processing (e.g., no convolution is used).*" But the training involves discriminator of LightGAN [3] that uses convolutional layers.

From Table 2, the number of train steps are directly proportional to the generation quality. Is it due to having a softmax over the key as discussed in Section 4.2 that forces slow update rules. In my understanding, and the authors' note, "*The first thing we observe is that for many steps, the “key” is almost one-hot (which is encouraged by the softmax), i.e., the part of the image is generated almost column-wise.*" the key is crucial in determining the attention or the generative patterns for the next iteration. Have the authors explored on other activations for key matrices.

Could the authors explain why there is more drastic effect of trains steps on the AFHQ dataset when compared to MetFaces dataset (as shown in Table 2). Have they explored the possible effects of the rank of an image on the generation process.



References:

1. Skorokhodov, Ivan, Savva Ignatyev, and Mohamed Elhoseiny. "Adversarial generation of continuous images." Proceedings of the IEEE/CVF Conference on Computer Vision and Pattern Recognition. 2021.
2. Gregor, Karol, et al. "Draw: A recurrent neural network for image generation." International conference on machine learning. PMLR, 2015.
3. Liu, Bingchen, et al. "Towards faster and stabilized gan training for high-fidelity few-shot image synthesis." International Conference on Learning Representations. 2020.
4. Zhu, Jiapeng, et al. "Low-rank subspaces in gans." Advances in Neural Information Processing Systems 34 (2021): 16648-16658.



**Summary Of The Paper:**

The work visualizes image generation as sequential low-rank approximations to the weight matrices in a FWP(Fast Weight Programming) setup. The motivation is to not generate high fidelity images but to rather enable human interpretable visualizations of the weight updates in a Neural Network (NN). The authors have shown human comprehensible weight update sequences in six popular datasets.



**Summary Of The Review:**

Overall, I appreciate the novelty in equating weight updation to image generation in FWP framework. The paper is clearly written and mostly easy to follow. My main concerns are with the limitations of the proposed approach in terms of its scalability w.r.t image size and to its applicability beyond the visualization of weight updates as discussed under the first point in the main review under methodology.

---

> ### Author Response · Authors · 2022-11-08
> **Response to Reviewer wMfX, part 1/2**
>
> We sincerely thank the reviewer for the invested time and effort, and useful comments, but we also have to be honest: due to both the structural and linguistic clarity issues, we generally found it hard to read and understand the main points of this review.
> Nevertheless, we did our best to interpret the reviewer's concerns, and
> hopefully addressed them all successfully.
> Please consider increasing the score if you find our response convincing.
>
> We'd like to start by resolving several misunderstandings.
>
> **Summary Of The Paper**
>
> > *... sequential low-rank approximations to the weight matrices in a FWP(Fast Weight Programming) setup.*
>
> Strictly speaking, the low-rank scenario is only studied in the ablation study in Sec. 4.3. In all other cases, *"we set the number of generation steps for FPAs to T = 64 ... such that the output images can be of full rank''* (as stated in Sec. 4.1.).
>
> > *The motivation is ... to rather enable human interpretable visualizations of the weight updates in a Neural Network (NN).*
>
> This is not accurate. We never state that our goal is to visualise the weight updates in NNs.
> We apply FWPs to image generation to demonstrate that the generic NN weight learning rules can effectively produce arbitrary complex patterns (we illustrate this through natural images), and to visualise these sequences of learning rules that produce images.
> This is stated in the introduction (fourth paragraph) and the conclusion.
> Please let us know if the reviewer still finds this confusing.
> In what follows, we'll refer to the motivation above as the *main goal/purpose* of this work.
>
> **Novelty and Originality**
>
> > *...one of the few works that draw similarities between images and weight matrices that are generated by another neural net. The recent of this type being the class of Implicit Neural Representations (INR) [1].*
>
> INRs and our work are clearly not of the same type.
> In [1], weights of the generator are modulated by the input noise, but the generator is the standard INR [wMfX, 6]; thus its image generation process itself does not leverage any NN weight generation techniques.
> In our work, it is the image generation process that is parameterised by classic learning rules conventionally used for NN weight generation.
> In fact, we do not really *"draw similarities between images and weight matrices that are generated by another neural net."*
> We simply treat images as arbitrary matrices, and generate them using a generic matrix generation process that is typically used to generate NN weight matrices.
> This may better be categorised as applying NN-weight matrix processing techniques to images (and vice versa).
> For example, we could have referred to [wMfX, 7] that applies the discrete cosine transform (typically used for images) to NN weight matrices.
>
> [wMfX, 6] Sitzmann et al. Implicit neural representations with periodic activation functions, NeurIPS 2020.
>
> [wMfX, 7] Koutnik et al. Evolving neural networks in compressed weight space, GECCO 2010.
>
> **Methodology**
>
> > *Given the poor generation quality of FPA, my question is if the performance vs interpretability/explainability tradeoff presented in the work is worth a shot.*
>
> We do not fully understand what is meant by *"worth a shot"* but in any case, please note that
> *"performance vs. interpretability tradeoff"* is not at all the motivation of our work.
> Please refer to the *main goal/purpose* above (described in the introduction/fourth paragraph and conclusion).
> Our goal is NOT at all to build an interpretable image generator.
>
> > *Can the method be adapted to exercise control over generations by manipulating the low-rank updates, ref [4] or restrict the low-rank updates to find discriminating components, ref [5], instead of a rank based updation approach.*
>
> Here the reviewer seems to propose some extensions or alternatives to our work (but we do not fully understand what exactly s/he is proposing; we also cannot find ref [5] in the review).
> Please note that the entire purpose of this work (the first paper on FPAs) is to study the generic NN weight learning rules without any further specifications.
> If the reviewer has more specific and concise questions, we'll be happy to answer them, but we leave any other extensions for future work.
>
> > *Why shouldn't one compare FPA with recurrent painting techniques like Draw [2].*
>
> We do not think such a comparison brings any useful information to support the main goal/claim of our work.
> If the reviewer thinks that such a comparison is needed, please concisely explain the reason.

---

> > ### Author Response · Authors · 2022-11-08
> > **Response to Reviewer wMfX, part 2/2**
> >
> > > *But the training involves discriminator of LightGAN [3] that uses convolutional layers.*
> >
> > Clearly, our statement *"without any explicit inductive bias"* refers to our generator architecture.
> > In the paper (second paragraph in Sec. 5) we even discuss that this mismatch between the convolutional discriminator and non-convolutional generator may be harmful.
> > (In passing, we note that the statement on inductive bias is anyway not crucial to the main purpose of this paper.)
> >
> > > *From Table 2, the number of train steps are directly proportional to the generation quality. Is it due to having a softmax over the key as discussed in Section 4.2 that forces slow update rules.*
> >
> > We suppose that by *"train steps"* the reviewer refers to the decoding/painting steps.
> > Please note that all visualisations
> > are done at test time; they are NOT visualisations of training steps.
> > Regarding softmax, yes, as we state in Sec. 4.2.,
> > softmax encourages the keys to be one-hot/sharp.
> > However, as we also discuss in Sec. 4.2., we do see that the model is capable of generating more "uniform" keys that cover more regions of the image where appropriate (e.g., to paint the background).
> >
> > > *Have the authors explored on other activations for key matrices.*
> >
> > No, we haven't. Please note that softmax is crucial for the stability of the delta rule in FWPs, which is not specific to our work.
> > Please see [wMfX, 8] for details: the delta rule in FWPs requires an activation function that produces positive elements that sum up to one.
> >
> > [wMfX, 8] Schlag et al. Linear Transformers Are Secretly Fast Weight Programmers, ICML 2021.
> >
> > > *Could the authors explain why there is more drastic effect of trains steps on the AFHQ dataset when compared to MetFaces dataset (as shown in Table 2). Have they explored the possible effects of the rank of an image on the generation process.*
> >
> > This is a valid question.
> > Since the metric we compute is FID (comparing the statistics of the generated images to those of the training sets),
> > there may be many reasons.
> > As the reviewer points out, the ranks of training images may have an impact: as we note in Sec 4.3., our model does exploit symmetries and regularities to produce these images within a limited number of steps.
> > However, this can not be the only reason, since these symmetries are also present in other animal faces in AFHQ Dog and Wild.
> > Diversity/variability of patterns across images should also have an impact on these results.
> > Overall, this is not an easy question with an obvious answer.
> >
> > > *But would appreciate a more vivid view of the FPA generator architecture and the role of a third NN to map z to initial hidden states of RNN layers of the slow NN.*
> >
> > We do not understand what is meant by *"a more vivid view of the FPA."*
> > Regarding the map from z to initial hidden states, it is just an architectural variation (starting the recurrent state from 0 vs. a random vector). Table 3 in the appendix specifies datasets on which this was useful.
> >
> > **Summary**
> >
> > > *My main concerns are with the limitations of the proposed approach in terms of its scalability w.r.t image size ...*
> >
> > We were surprised to find a new main concern in the summary: *"limitations of the proposed approach in terms of its scalability w.r.t image size."*
> > Nevertheless, the question on scalability is a valid question also asked by Reviewers 81GC and oERk.
> > Please note that the difficulty of scaling image generation methods to higher resolutions is not specific to our method.
> > Even for the conventional convolution-based GAN models, this has been a research question in multiple works (see e.g., [wMfX, 9] and [wMfX, 10]).
> > Similarly, we feel that this is not something that we can easily fix in this first paper on FPAs.
> > If the reviewer wishes, we'll be happy add more comments on this in the revision but please note that the current text already does mention this limitation explicitly by citing [wMfX, 10].
> >
> > [wMfX, 9] Odena et al. Conditional Image Synthesis with Auxiliary Classifier GANs. ICML 2017
> >
> > [wMfX, 10] Karras et al. Progressive Growing of GANs for Improved Quality, Stability, and Variation. ICLR 2018
> >
> > > *... and to its applicability beyond the visualization of weight updates as discussed under the first point in the main review under methodology.*
> >
> > We are not sure which points in the main review the reviewer is referring to regarding the issue of *"applicability beyond the visualization of weight updates"* but since we replied to all questions in the corresponding section, we assume that the reviewer can find the corresponding answer.
> >
> > Overall, we believe we thoroughly addressed all concerns raised by the reviewer.
> > We also hopefully resolved many misunderstandings.
> > If you find our response convincing, please consider increasing the score.
> > Otherwise, please concisely tell us your reasons for rating our work as 5 (voting for rejection).
> > Thank you.

---

> > > ### Author Response · Authors · 2022-11-17
> > > **Friendly reminder**
> > >
> > > This is a friendly reminder that the discussion period ends tomorrow.
> > > If you find our response convincing, please consider increasing the score.
> > > Thank you!

---

### Author Response · Authors · 2022-11-08
**Global Response**

We thank all the reviewers for their valuable feedback and comments.
We are glad to see that all reviewers found our work original/interesting and well written.
We believe that we have good answers to all main concerns raised by the reviewers.
Please find our individual responses below.
If you find them convincing, please consider increasing the score.

For the moment, we will keep the originally submitted PDF as is, such that we can refer to the originally submitted/reviewed text (unless any of the reviewers prefers the PDF to be updated already).
We will apply all promised edits in the rebuttal if the paper is accepted.

---

### Decision · Program_Chairs · 2023-01-20

**Decision:**

Accept: poster

**Justification For Why Not Higher Score:**

The paper essentially shows that fast weight programmers can be used for image generation, but does not give any clear reasons why it should be preferred over existing algorithms, or show that it has unique properties with regards to other approaches. It also does not give any clear insight into fast weight programmers more generally, or give insights that could be used when applying this approach on other problems.

**Justification For Why Not Lower Score:**

A majority of reviewers felt the paper should be accepted.

**Metareview: Summary, Strengths And Weaknesses:**

The paper presents an application of fast weight programmers to image generation tasks, termed fast weight painters (FPAs). FPAs are trained in a GAN framework on a variety of image generation datasets. Although FPAs do not achieve state-of-the-art performance on image generation, the key result of the paper is to demonstrate that this generic learning rule can produce reasonable results on this task.

Initially this paper received borderline reviews. Reviewers understood that the goal of the paper was not to achieve state-of-the-art image generation, and generally found the paper interesting, and a novel approach to image generation. Reviewers raised concerns with many details of the paper which were mostly resolved by the author’s responses, and after the rebuttal period the reviewer scores were generally positive.

The AC met with Reviewers oERk and 81gc to discuss the paper in more detail. Both reviewers found the approach novel and interesting, but had some borderline feelings about the paper. Overall, they felt unclear about the positioning of the paper, following the discussion that Reviewer oERk had with the authors during the discussion phase. The paper demonstrates that FPA can be used for image generation, but it does not outperform existing image generation methods. Per the author’s discussion with Reviewer wMfX, the proposed algorithm is also not intended to trade off image generation performance vs interpretability. Overall, the reviewers could not see any clear advantage of the proposed algorithm over existing methods for image generation, apart from using a different architecture. It is also not clear whether the results of the paper offer insight into fast weight algorithms that could be applied to other tasks; indeed in discussion with Reviewer oERk the authors clarify that they “do not intend to draw any generic conclusion” and “can not tell whether this observation is helpful for future studies.”

The AC agrees with the sentiments raised by Reviewers oERk and 81gc. The paper would be much more convincing if the authors could either (1) give some clear reason why sequential learning might be preferred to existing image generation algorithms; or (2) use this as a case study to bring some new insights into sequential learning rules generally. For (1), FPA need not outperform existing methods, or generate high-resolution images; but there should be some argument or reason why this algorithm is interesting for image generation, other than its novelty.

However due to the novel and interesting approach to image generation, the AC believes that the submission would be appreciated by the ICLR community. The authors are encouraged to revise their paper for the camera-ready version, incorporating feedback from the reviewers to clarify and improve the positioning of the paper.

**Note From Pc:**

if the above contains the word "oral" or "spotlight" please see: "oral" presentation means -> notable-top-5% and "spotlight" means -> notable-top-25%. As stated in our emails, we are disassociating presentation type from AC recommendations

**Summary Of Ac-Reviewer Meeting:**

The AC met with Reviewers oERk and 81gc to discuss the paper. We discussed that the method is interesting, but that it does not have any clear advantages for image generation. We also discussed the lack of high-resolution outputs, but this was not a major point of concern for reviewers. We primarily discussed the overall takeaways that the community might have from this paper: are there any new insights either for image generation or for fast weight algorithms? We could not identify any clear takeaways along either lines; although the paper is interesting, it could benefit from some minor revision to improve its positioning.